# Effect of Nickel Addition on Solidification Microstructure and Tensile Properties of Cast 7075 Aluminum Alloy

Kai Wang [1],*, Haoran Qi [1], Simu Ma [1], Linrui Wang [1], Naijun He [1],* and Fuguo Li [2]

1 College of Materials Science and Engineering, Chongqing University, Chongqing 400030, China; qihaoran77@sina.com (H.Q.); msm@stu.cqu.edu.cn (S.M.); 18300851798@139.com (L.W.)
2 College of Materials Science and Engineering, Northwest Polytechnical University, Xi'an 710072, China; fuguolx@nwpu.edu.cn
* Correspondence: wangkai@cqu.edu.cn (K.W.); nj.he@163.com (N.H.)

**Abstract:** In order to explore the casting technology of a high–strength aluminum alloy, the effects of nickel on the solidified microstructure and tensile properties of a 7075 aluminum alloy were studied. 7075 aluminum alloys without nickel and with 0.6% and 1.2% nickel were prepared by a casting method. The results showed that the increase of Ni content in the 7075 alloys increased the liquidus temperatures, primary $\alpha$ (Al) grains were refined significantly, and the divorced eutectic structure was gradually formed among $\alpha$ (Al) grains with the preformation of the $Al_3Ni$ phase. In comparison, the 7075 alloy with 0.6% nickel content had less intergranular shrinkage porosity, and its elongation and ultimate tensile strength was enhanced 45% and 105% higher than those of the as-cast 7075 aluminum alloy, respectively. When the Ni content was increased to 1.2%, the eutectic phases of the alloy became much coarser compared to the other two alloys, and the mechanical properties obviously reduced too.

**Keywords:** 7075 aluminum alloy; eutectic solidification; Al–Ni alloy; intergranular structure; mechanical property

## 1. Introduction

To achieve climate neutrality, it is urgent to reduce transport emissions and increasingly develop lightweight technologies. Aluminum alloys are the best choice as a lightweight material in terms of weight reduction effect and price [1]. It is well known that Al–Zn–Mg–Cu alloys have high strength, excellent fracture toughness, and good corrosion resistance, and they are widely used in aerospace areas [2–7]. However, Al–Zn–Mg–Cu alloys usually show a high cast cracking tendency and poor mold-filling behavior [8,9]. Traditionally, these alloys were used for manufacturing using plastic deformation methods, which caused high manufacturing cost and long production cycle. In order to achieve low-cost manufacturing of high-performance complex structural parts, many researchers tried to improve the hot cracking resistance of these high-strength aluminum alloys. On the one hand, the mechanical properties of traditionally cast aluminum alloys were enhanced by alloying technologies [6,9]. On the other hand, high-strength alloys can also be manufactured by special casting methods such as controlled diffusion solidification [10].

Alloying is an efficient method to improve the comprehensive properties of Al–Zn–Mg–Cu alloys [11,12]. The addition of Zr and Sc to Al–Zn–Mg–Cu alloys can promote the formation of equiaxed grains and coherent $Al_3(Zr,Sc)$ particles in solidification structure, which led to improved mechanical properties [12–15] and a superior hydrogen embrittlement resistance [16]. It was well known that both equiaxed grains and large eutectic amount in solidification structure can reduce the pressure drop of the intergranular channel and subsequently improve the intergranular feeding ability [17]. When the grain morphology changed from developed dendrite to equiaxed in solidification structure [10,12], it caused a

decrease in the dendrite coherency point (DCP) temperature and an improvement in intergranular feeding ability. The research on eutectic solidification of AA7068 aluminum alloys showed that the residual liquid phase tended to attach to the $Mg(ZnCuAl)_2$ phase to form a lamellar eutectic structure [18], while both the inhibition of $\alpha(Al)$ secondary nucleation and the promotion of divorced eutectic tended to reduce the hot cracking tendency of the alloy. It was generally believed that eutectic composition solidifies had lower DCP temperature, better heat crack resistance, and flow ability [19]. Therefore, some researchers added small amounts of elements to control eutectic solidification behavior of the Al–Zn–Mg–Cu alloy to enhance feeding ability. During the non-equilibrium solidification, the residual liquid phase of nominally single-phase Al–Zn–Mg–Cu aluminum alloy in the last stage of solidification formed eutectic phases [18]. Obviously, eutectic nucleation and growth during solidification also affects the pressure drop of the intergranular feeding channel in the solidified alloy. The addition of nickel in alloys can reduce the thermal expansion coefficient of the alloy, improve the heat resistance, and eliminate the harmful effects of impurities iron in aluminum alloys [20]. Usually, nickel element forms an $Al–Al_3Ni$ eutectic phase in aluminum alloy [12,21], and the $Al_3Ni$ compound has low solid solubility in the matrix and high thermal stability, which can prevent dislocation movement and grain boundary slip and effectively improve the high temperature mechanical strength [22]. Previous studies showed that the addition of Ni to Al–Zn–Mg–Cu (7xxxx) alloy reduced the intergranular shrinkage defects and the hot cracking tendency. A reasonable amount of nickel addition in the Al–Zn–Mg–Cu alloy can also improve its strength, elongation, and other properties comparatively [6,23]. However, tensile fracture of the alloy was mainly composed of grain boundary cracks. It was analyzed that intergranular discontinuities may be connected with the eutectic structure during solidification, but the interaction between solidification microstructure and mechanical properties is still lacking.

In order to develop cast high strength alloys, the eutectic solidification behavior and its effects on mechanical properties should be studied more. In this study, the influence of nickel element on the solidification structure of a 7075 aluminum alloy was studied first, then the non-equilibrium solidification behavior and solidification microstructural evolution were discussed. Finally, the effects of nickel addition on the microstructure and tensile strength of a 7075 aluminum alloy were described.

## 2. Experimental Materials and Methods

Due to a large solidification temperature range and high-strength properties, the 7075 alloy was chosen for this study. The raw materials for the casting process were 7075 aluminum alloy ingots and Al–20 wt% Ni master alloy. Based on previous research results [6], casting samples of 7075 aluminum alloy without nickel and with 0.6 wt% and 1.2 wt% nickel by weight percentage were prepared. The 7075 aluminum alloy with additional 0.6% and 1.2 wt% nickel is abbreviated as 7075 Al–0.6 Ni and 7075 Al–1.2 Ni hereafter. The three types of aluminum alloy samples were prepared by the same melt casting processes. Table 1 shows the chemical composition of the three types of aluminum alloys.

**Table 1.** Chemical composition of aluminum alloy samples (Wt%).

| Alloy | Elements | | | | | | | |
|---|---|---|---|---|---|---|---|---|
| | **Zn** | **Mg** | **Cu** | **Cr** | **Si** | **Fe** | **Ni** | **Al** |
| 7075 Al | 5.6 | 2.5 | 1.6 | 0.23 | 0.4 | 0.5 | 0 | Bal. |
| 7075 Al–0.6 Ni | 5.32 | 2.37 | 1.54 | 0.22 | 0.38 | 0.47 | 0.6 | Bal. |
| 7075 Al–1.2 Ni | 5.06 | 2.26 | 1.45 | 0.21 | 0.37 | 0.46 | 1.2 | Bal. |

These alloys ingots were produced by the following procedures. Firstly, 7075 aluminum ingots were melted in a graphite crucible at 760 °C by using a 3.5 KW electric resistance furnace (SG-G10123, Tianjin Zhonghuan Electric Furnace Co., LTD, Tianjin,

China). Secondly, the Al–20 wt% Ni master alloy was added into 7075 alloy melts contained in the graphite crucible to achieve 7075 Al–0.6 Ni and 7075 Al–1.2 Ni alloys, and then the alloy melt was heated to 780 °C. After 30 min holding time, the melt was refined at this temperature by using hexachloroethane ($C_2Cl_6$). After that, the Al–5Ti–B alloy with 0.3% mass ratio was added to the molten aluminum to refine grains while stirring thoroughly and maintaining the temperature for 30 min to ensure the homogeneity of the composition. Finally, the alloy melt was held at 735 ± 5 °C for 15 min, and then it was poured into a cylindrical steel mold and a triangular steel mold preheated to 200 °C, respectively. The cylindrical mold with dimensions of Φ40 mm × 60 mm was used for the investigation of the solidification behavior, and the triangular mold together with a gating and feeder system was used to produce triangular cast ingots for the preparation of tensile specimens. The dimensions of the triangular cast ingot were 60 mm in length, 30 mm in height, and 25 mm in thickness.

In order to achieve the cooling curves during solidification of these alloys, a K-type thermocouple (abbreviated as TC) was fixed in the center of the cylindrical mold cavity, as shown in Figure 1. During solidification, temperature data were recorded every 100 ms using high-speed data acquisition equipment, and the cooling curves were plotted in Origin software. Important parameters in liquidus and eutectic nucleation regions were calculated using the first derivative cooling curves. Samples used for metallographic preparation were sectioned from the region surrounding the thermocouple tip for these cast billets.

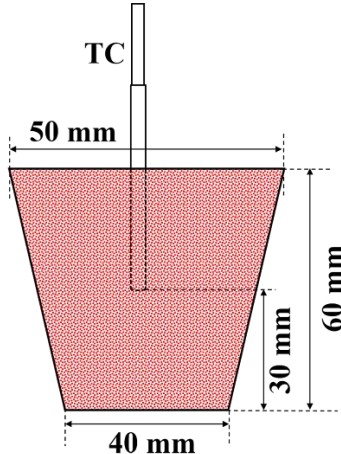

**Figure 1.** Dimensions of the steel mold used for the measurement of cooling curves.

Tensile test specimens were extracted from these cast ingots produced through the triangular mold, and the gauge length, width, and thickness of the static tensile specimen were 13 mm, 4 mm, 2.5 mm, respectively. According to ASTM E8/E8M [24], the tensile properties of the cast alloys were conducted at a crosshead speed of 1 mm/min on a universal testing machine. Elongation of these specimens was obtained by fitting the specimen back together after facture and measuring the change in length. The tensile tests were carried out three times for each of these cast aluminum alloys. After the metallographic samples were cut from the cast billets, these metallographic samples were ground, polished, and etched in Keller solution (2.5% $HNO_3$ + 1.5% HCL + 1% HF + 95% Distilled water). The metallographic microstructures were observed by optical microscope (ZEISS Axioskop 40, ZEISS Group, Göttingen, Germany), and the primary phase size or the grain boundary intersection of the solidified structure was measured by a line intercept method according to ASTM E112-12 [25]. The detailed microstructures were detected by using a scanning electron microscope (SEM) equipped with EDS (Hitachi TM4000plus, Hitachi High-Tech Corporation, Tokyo, Japan) and an X-ray diffractometer (XRD, Empyrean, Malvern Panalytical, Almelo, The Netherlands). An electron probe microanalysis (EPMA, JXA-8530F plus, JEOL Ltd., Tokyo, Japan) was used to analyze the element distribution of the different alloys. In addition, PanPhaseDiagram, which is a phase diagram calculation module of Pandat

software, was used to calculate the multicomponent phase equilibria and precipitation sequence of these alloys.

## 3. Results

### 3.1. Microstructure

The optical microstructures of the as-cast alloys are shown in Figure 2, in which the white primary $\alpha$–Al phases were cellular dendrites, and microstructure in dark grey contrast between primary grains was considered as intermetallic phases and eutectic phases. The grain boundary intersection of the cast 7075 alloy was determined as $58 \pm 4.1$ μm. Comparing with the solidification microstructure of the 7075 aluminum alloy (Figure 2a), the primary grain size of the Ni–containing 7075 aluminum alloys decreased significantly due to the addition of nickel, while the distribution of primary grains was more uniform. Statistically, the grain boundary intersection of the 7075 Al–0.6 Ni alloy and the 7075 Al–1.2 Ni alloy was determined as $50 \pm 6.2$ μm and $44 \pm 6.8$ um, respectively. Once the mass fraction of Ni increased from 0.6% to 1.2% (Figure 2b,c), more equiaxed primary grains were also observed in the 7075 Al–1.2 Ni alloy (Figure 2c).

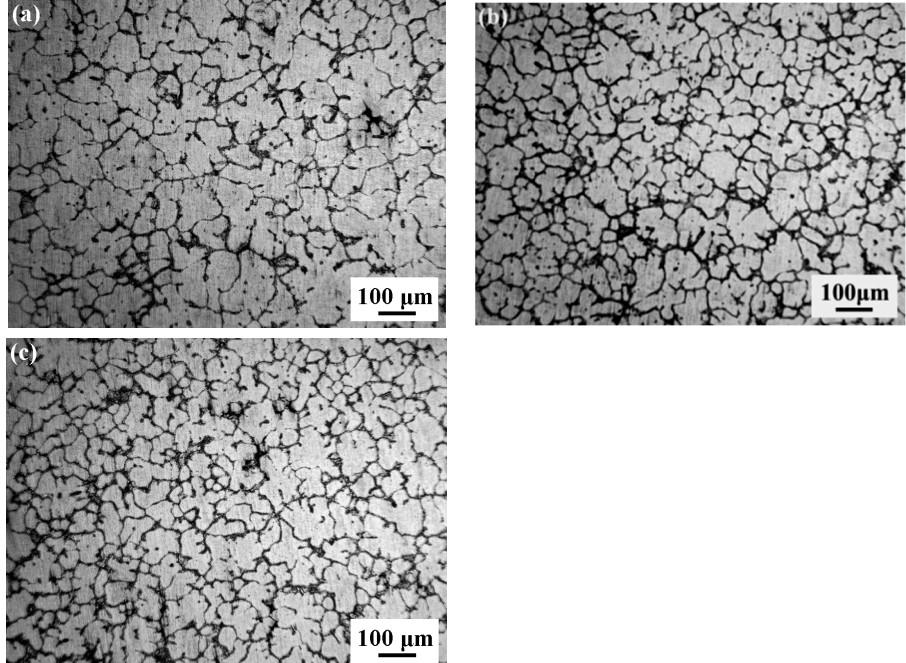

**Figure 2.** Optical microstructure of the 7075 aluminum alloy with different Ni content, (**a**) 7075 Al, (**b**) 7075 Al–0.6 Ni, (**c**) 7075 Al–1.2 Ni.

Figure 3 shows the XRD patterns obtained from these three alloys. In addition to $\alpha$–Al and MgZg$_2$ in these three alloys, the Al$_3$Ni compound phase was found in the Ni–containing 7075 alloys due to the exothermic reaction between Al and added Ni in 7075 aluminum alloy. These results were consistent with the previous research [6,20], indicating that the addition of Ni in the 7075 aluminum alloy did not cause changes in other phases in the alloy. Based on the Al–Ni phase diagram, it has been known that several intermetallic compounds (IMCs) including Al$_3$Ni, Al$_3$Ni$_2$, Al$_4$Ni$_3$, AlNi, Al$_3$Ni$_5$, and AlNi$_3$ can exist in an Al–Ni system compound depending on the nickel content in this alloy, and the formation of Al$_3$Ni in these Ni–containing 7075 aluminum alloys can be attributed to a low Ni content of 1.2% in the 7075 alloy.

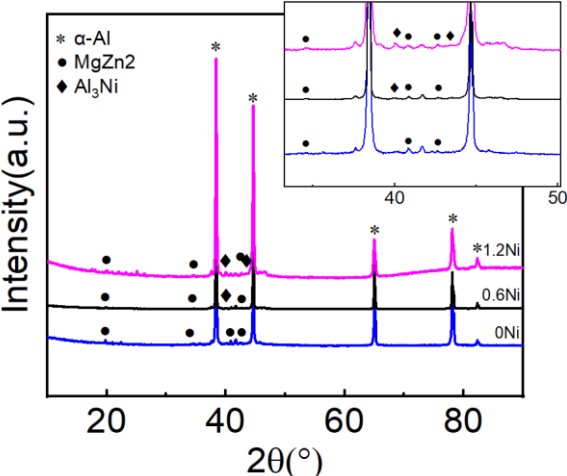

**Figure 3.** XRD analysis of as-cast 7075 aluminum alloys with different Ni contents.

Figure 4 shows the intergranular microstructure of these three alloys, showing eutectic phases, discontinuous defects (marked by the white arrows), and a small amount of shrinkage porosity (marked by white circles), mainly distributed among α–Al primary grains. With the increase of nickel content in the 7075 aluminum alloy, the amount of intergranular phases increased obviously. It was worth noting that some differences including porosity and crack can be observed in the 7075 aluminum alloy, and that there were more porosities at triple grain boundaries in the 7075 Al–1.2 Ni alloy than in the other two alloys.

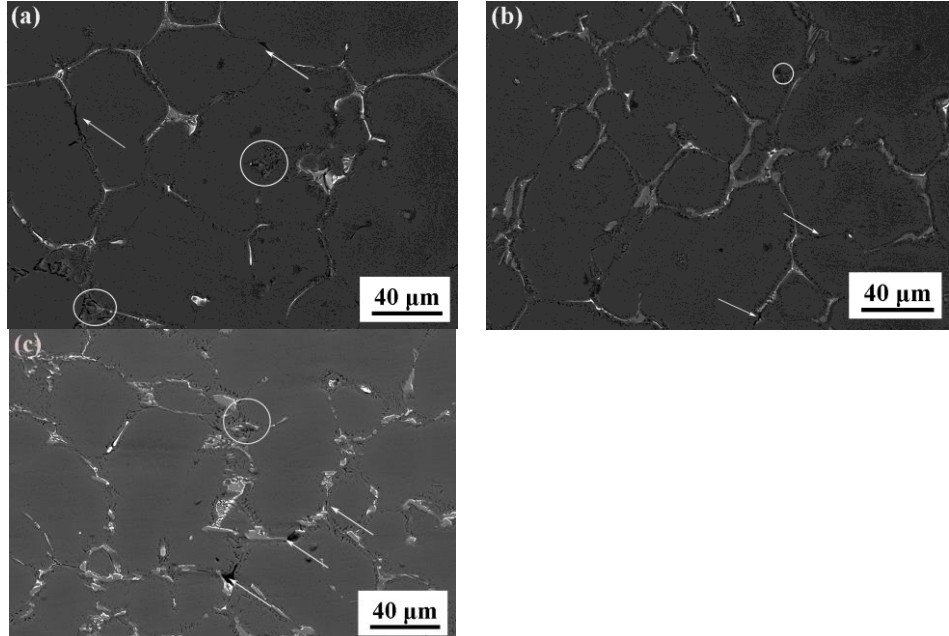

**Figure 4.** SEM images showing the cast microstructure of three alloys, (**a**) 7075 Al, (**b**) 7075 Al–0.6 Ni, (**c**) 7075 Al–1.2 Ni.

Table 2 shows the EDS results of the labeled regions in Figure 4. According to previous research [6,18], a regular lamellar eutectic structure was formed in the 7075 Al alloy, which consisted of alternate grey α–Al and bright MgZn$_2$ (η phase) lamellae (Figure 5a). It has been reported that Mg(Zn, Al, Cu)$_2$ (σ phase) and Al$_2$CuMg (S phase) are common in 7xxx series aluminum alloys with low Cu: Mg ratio [26,27]. The EDS results shown in Table 2 indicate that the phase marked at B was S phase [24]. In addition, the EDS results shown in

Table 2 indicate that the phase marked at C contained much higher Fe (10.63 wt%) than any other areas, and Cu in the 7075 alloy chemically combined with Al and Fe to form tetragonal $Al_7Cu_2Fe$ constituent particles during solidification [28]. Therefore, Fe–containing phase in those alloys may be $Al_7Cu_2Fe$ intermetallic, and the intermetallic compound $Al_7Cu_2Fe$ mixed with $Al_3Ni$ in the Ni–containing 7075 aluminum alloys. The lamellar eutectic among $\alpha$–Al primary phases in the 7075 Al alloy was predominantly replaced by dispersed block or striped coarser and blocked $MgZn_2$, $Al_2CuMg$, $Al_3Ni$, and $Al_7Cu_2Fe$ particles in the Ni–containing 7075 Al alloys, which dispersed among grains and was regarded as divorced eutectic phases (Figure 5b,c). With the increase of nickel content in the 7075 Al alloy, the amounts of intergranular phases increased, but the amount of $Al_2CuMg$ phase did not decrease significantly. The $\alpha$–Al primary grain refining in the Ni–containing 7075 alloys was related to the pinning effects, in which a large number of intergranular compounds at grain boundaries hindered the movement of the grain boundaries.

**Table 2.** Chemical composition of the labeled regions in Figure 5, (wt%).

| Point | Zn | Mg | Cu | Si | Fe | Ni | Al |
|---|---|---|---|---|---|---|---|
| A | 2.41 | 1.56 | 0.46 | 0 | 0 | 0 | 95.55 |
| B | 3.56 | 1.43 | 27.64 | 0.25 | 0.05 | 0 | 67.02 |
| C | 1.92 | 0.84 | 5.14 | 2.56 | 10.63 | 0 | 78.09 |
| D | 1.73 | 0.69 | 2.03 | 0.26 | 4.45 | 7.18 | 83.04 |

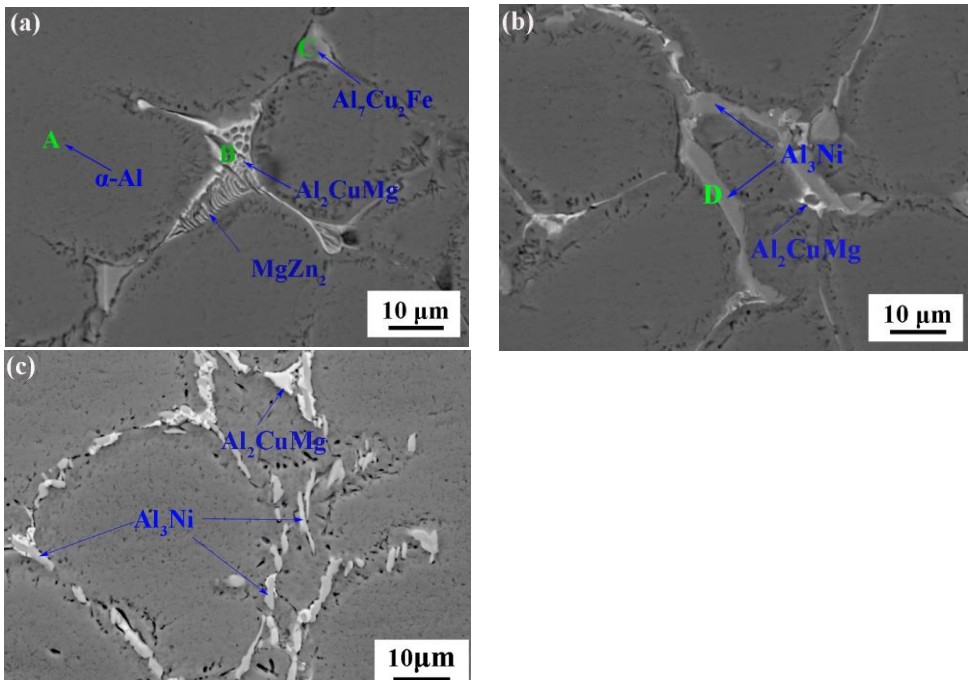

**Figure 5.** SEM images of three alloys (**a**) 7075 Al, (**b**) 7075 Al–0.6 Ni, (**c**) 7075 Al–1.2 Ni.

In order to identify the characteristics of intergranular phases further, EPMA technology was used to detect the distribution of alloy elements. The EPMA results of 7075 Al, 7075 Al–0.6 Ni, and 7075 Al–1.2 Ni alloys are shown in Figures 6–8, respectively, and the following phenomena can be found: (1) Mg and Zn were mainly distributed inside the $\alpha$–Al primary grains; it was believed that most of Mg and Zn dissolved in Al Matrix; (2) Cu, Fe, and other elements were mainly concentrated at the intergranular areas, and there were much more Fe elements at the intergranular areas in the as-cast 7075 alloy (Figure 6); (3) there was obvious Cu segregation to the grain boundaries with the dissolution of Mg, which verified that $Al_2CuMg$ phases formed in these alloys [27]; (4) in the Ni–containing

7075 aluminum alloys shown in Figures 7 and 8, the Ni element was also mainly segregated at the primary grain boundaries, and it was close to the aggregation areas of the Fe element.

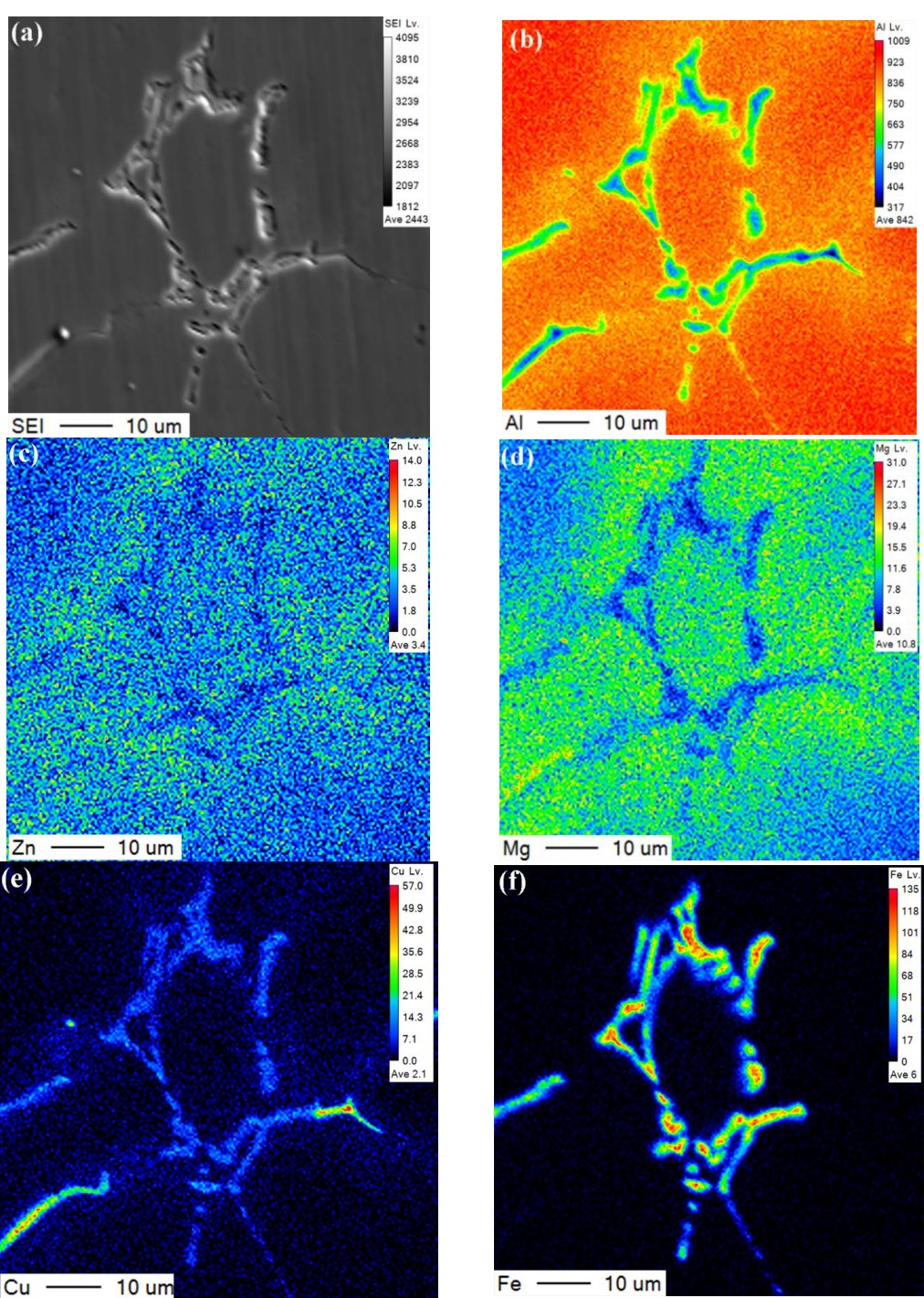

**Figure 6.** Secondary electron image and elemental plane distribution of as-cast 7075 alloy by EPMA. (**a**)—Secondary electron image, (**b**)—Al, (**c**)—Zn, (**d**)—Mg, (**e**)—Cu, (**f**)—Fe.

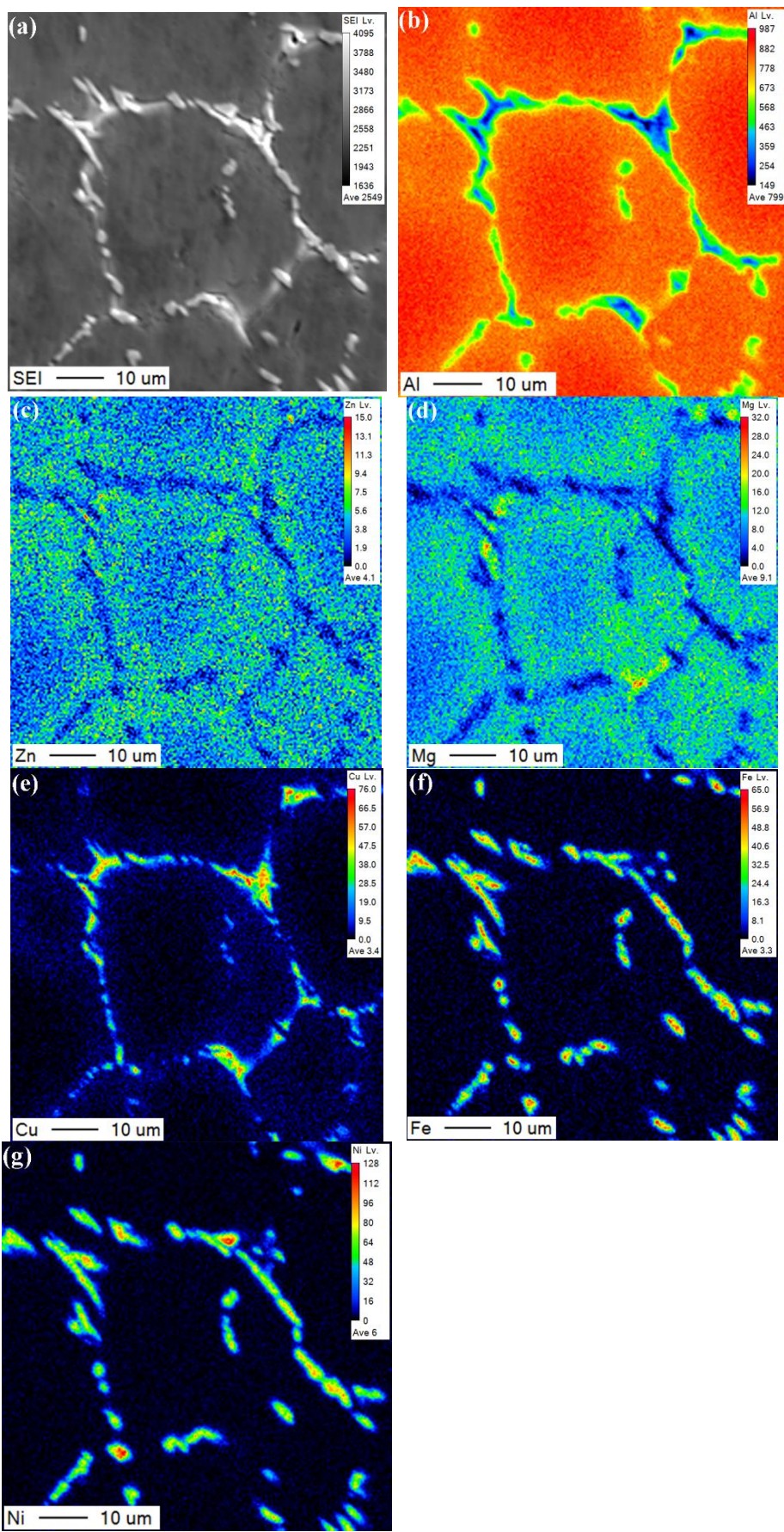

**Figure 7.** Secondary electron image and elemental plane distribution of the as-cast 7075 Al–0.6 Ni by EPMA. (**a**)—Secondary electron image, (**c**)—Zn, (**d**)—Mg, (**e**)—Cu, (**f**)—Fe, (**g**)—Ni.

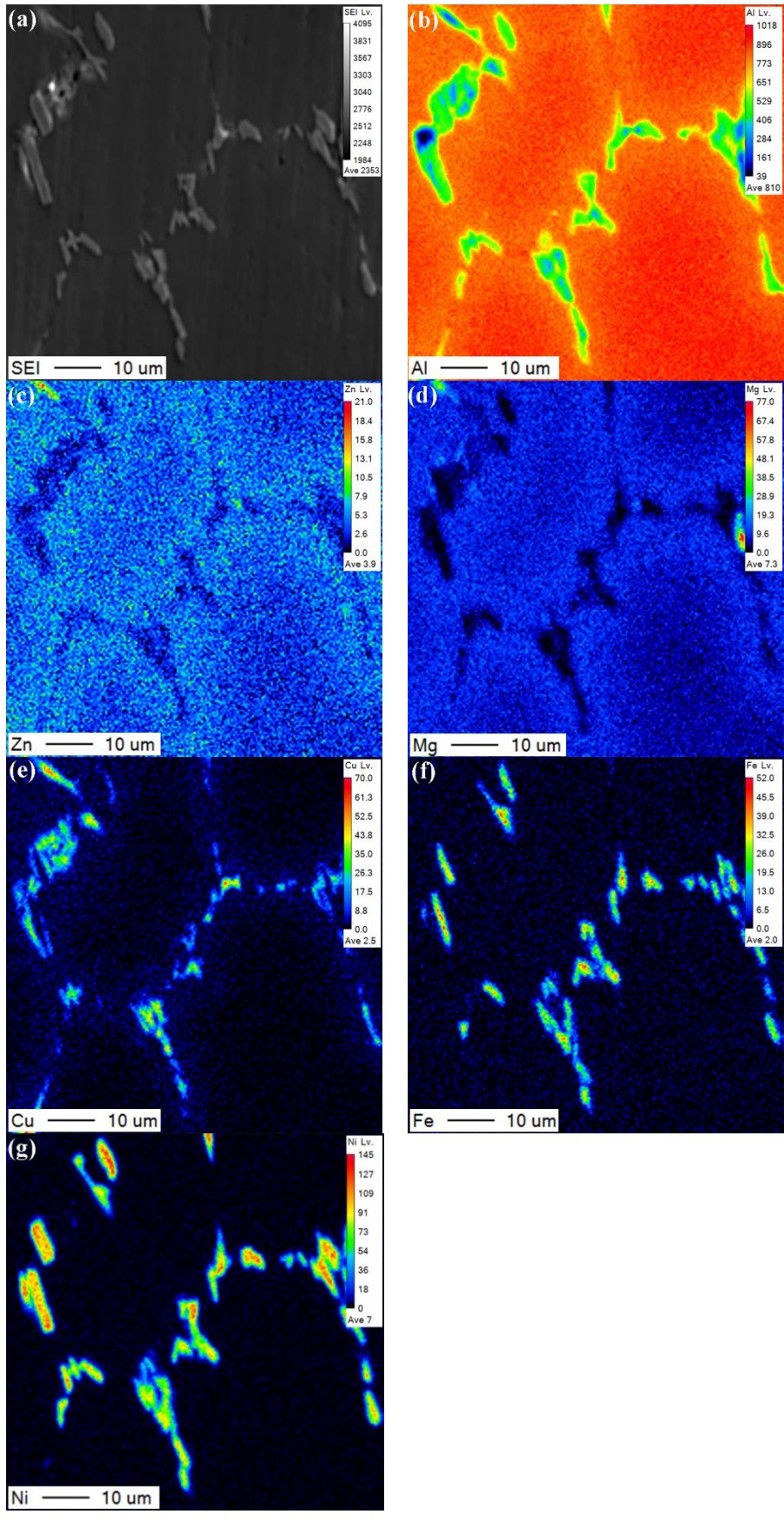

**Figure 8.** Secondary electron image and elemental plane distribution of as-cast 7075 Al–1.2 Ni by EPMA. (**a**)—Secondary electron image, (**b**)—Al, (**c**)—Zn, (**d**)—Mg, (**e**)—Cu, (**f**)—Fe, (**g**)—Ni.

Figure 9 shows the measured cooling curves of these three alloys and the calculated first derivative curve obtained from the cooling curves. During primary phase precipitation, a large amount of latent heat was released, which caused a rapid drop in temperature. In the process of secondary phase precipitation, a small amount of latent heat was released, thus a solid-state region occurred. Derivation curves allowed to identify the phase changes of the alloys that happened during the solidification period. According to previous research on solidification behavior [29–31], alloy solidification starts from the nucleation of primary grains, and the nucleation temperature was determined as $T_{N,\alpha}$. The nucleation temperatures, solidus temperatures, and eutectic reaction temperatures of the experimental alloys are shown in Table 3. It can be seen that the nucleation temperature and eutectic reaction temperature of the 7075 alloy increased with the increase of Ni content in the alloy.

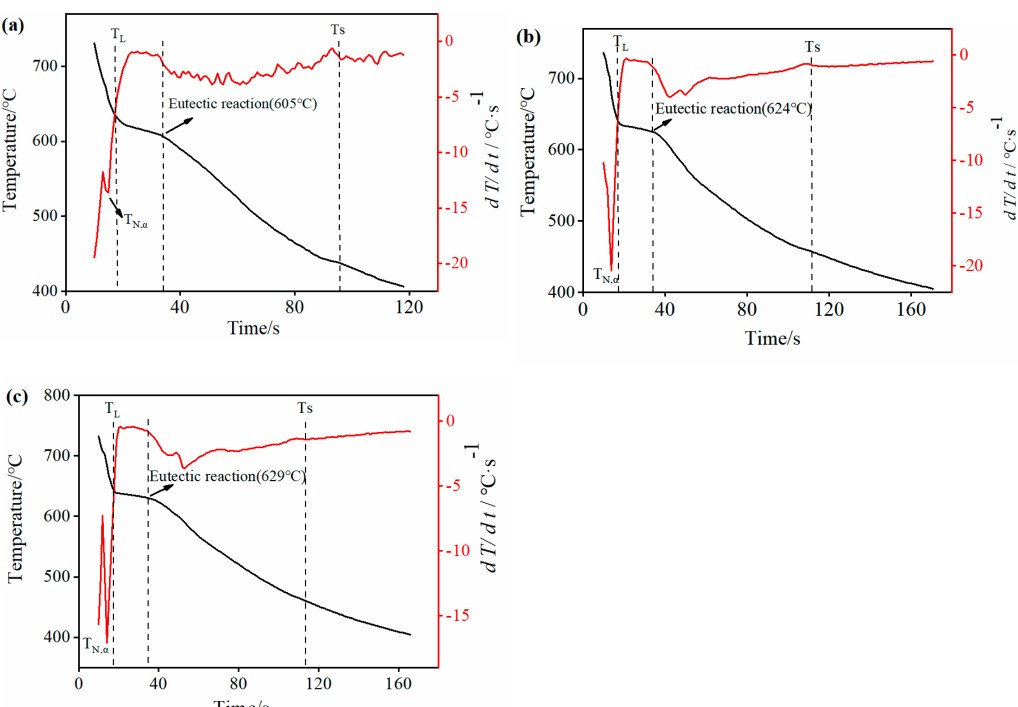

**Figure 9.** Cooling curve and its first derivative curve, with major points labeled for three alloys, (**a**) 7075 Al, (**b**) 7075 Al–0.6 Ni, (**c**) 7075 Al–1.2 Ni.

**Table 3.** Initial nucleation temperatures and eutectic reaction peak temperatures of these three alloys, (°C).

| Alloy | $T_{N,\alpha}$, Nucleation Temperature | Ts, Solidus Temperature | Eutectic Reaction |
|---|---|---|---|
| 7075 Al | 633 | 440 | 605 |
| 7075 Al–0.6 Ni | 635 | 457 | 625 |
| 7075 Al–1.2 Ni | 640 | 459 | 629 |

*3.2. Mechanical Property*

Figure 10 shows the ultimate tensile strength and elongation of 7075 Al, 7075 Al–0.6 Ni, and 7075 Al–1.2 Ni alloys at room temperature. The strength and elongation of the as-cast 7075 alloy were determined as 144 MPa 0.95%, respectively. When the mass fraction of Ni in the 7075 alloy was 0.6%, the ultimate strength and elongation of the 7075 Al–0.6 Ni alloy increased to 209 MPa and 1.95%, respectively. When the mass fraction of Ni was 1.2%, the ultimate tensile strength of the 7075 Al–1.2 Ni alloy decreased to 187 MPa, although the elongation of the sample increased a little compared with that of the 7075 Al–0.6 Ni alloy.

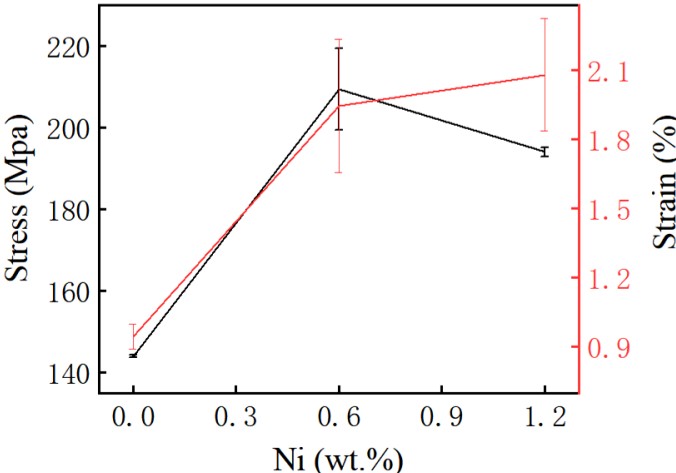

**Figure 10.** Tensile mechanical properties of as-cast alloys at room temperature (ultimate tensile strength and elongation).

Figure 11 shows the fracture appearance of the studied alloys, and the overall fracture surface was perpendicular to the tensile axis. The fracture surface of the 7075 Al alloy presented intergranular shrinkage porosity and separated $\alpha$(Al) grains, exhibiting brittle intergranular failure. It should be noted that the microstructure of the studied alloys also showed many intergranular shrinkage porosities and other discontinuities (Figure 4). In contrast, a few separated $\alpha$(Al) grains, some cleavage planes, and slight intergranular shrinkage porosities can be observed on the fracture surface of the 7075 Al–0.6 Ni alloy, which is believed is a mixed mode fracture. However, there were much more intergranular shrinkage porosities and less cleavage planes in the 7075 Al–1.2 Ni alloy compared with these in the 7075 Al–0.6 Ni alloy.

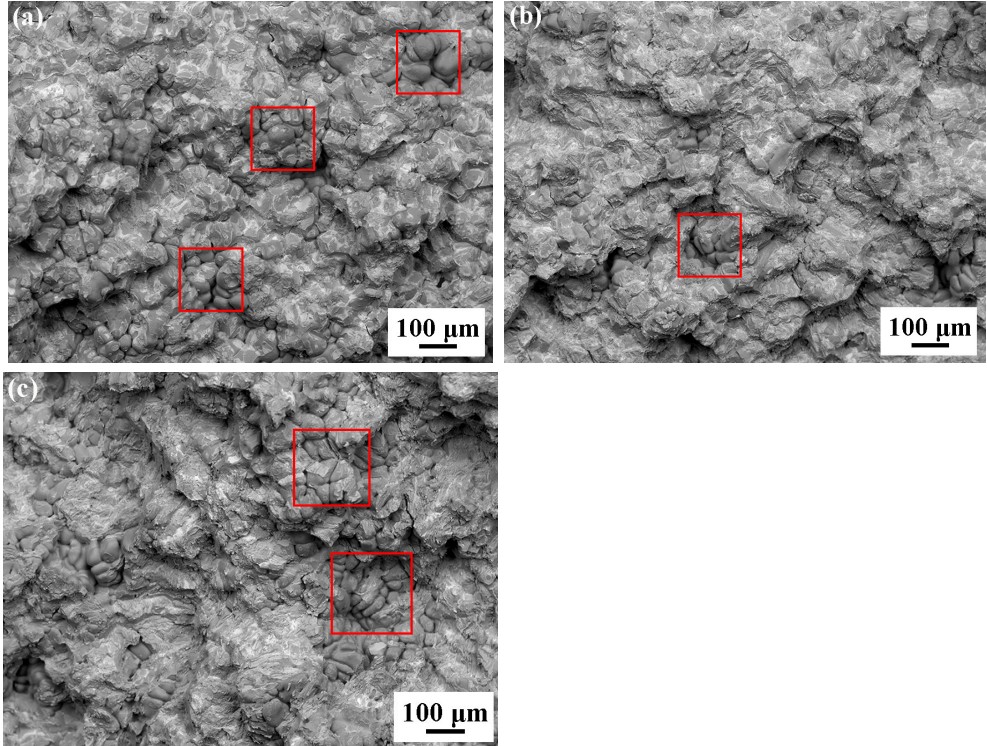

**Figure 11.** SEM fractographs of the fracture surface for alloy tensile samples, (**a**) 7075 Al, (**b**) 7075 Al–0.6 Ni, (**c**) 7075 Al–1.2 Ni.

## 4. Discussion

The results showed that the addition of nickel in 7075 aluminum alloy changed the morphology of the primary phase and intergranular structure significantly, so that the mechanical properties also changed greatly. The change of as-cast microstructure also depended on the solidification behavior of the alloys.

### 4.1. Microstructure Characteristics of Studied Alloys

In the investigation, a commercial 7075 aluminum alloy was used, in which the Fe element in the 7075 Al alloy mainly formed $Al_7Cu_2Fe$, while the Cu element in this alloy mainly formed $Al_2CuMg$. Compared with the 7075 aluminum alloy, only $Al_3Ni$ compound was found in the Ni–containing 7075 alloys due to the low addition levels of nickel, which is consistent with the reported research [21]. $Al_7Cu_2Fe$, $Al_2CuMg$, and$MgZn_2$ together with $Al_3Ni$ formed as intergranular phases in 7075 Al–0.6 Ni and 7075 Al–1.2 Ni alloys. Because of limited solubility of Ni in solid Al, Ni may form IMCs in aluminum alloys even at its low contents, such as $Al_3Ni$, $Al_3Ni_2$, etc. [32]. Furthermore, the Al–$Al_3Ni$ eutectic structure can be formed in the aluminum nickel eutectic alloy [33]. According to the microstructural results, when the Ni content in the 7075 alloy increased from 0.6% to 1.2%, the amount of $Al_3Ni$ intermetallic compounds increased, and its distribution among primary grains became more intensive. According to existing research, the amount of $MgZn_2$ and $Al_2CuMg$ in Ni–containing 7075 alloys stayed constant [21].

Combining the microstructures shown in Figure 5 with the given element distribution shown in Figures 6–8, it can be found that the $Al_2CuMg$ phase in the 7075 aluminum alloy and the eutectic Al form the coupled eutectic structure with lamellar morphology, and the $Al_7Cu_2Fe$ phase was a large block. In the Ni–containing 7075 alloy, intermetallic compounds (IMCs) $Al_7Cu_2Fe$ and $Al_3Ni$ were in coarse ribbons, and $Al_2CuMg$ and $MgZn_2$ were scattered among them to form an intergranular mixture, which was considered as a kind of divorced eutectic [21].

The refined grains in all three alloys can be attributed to the following reasons. Firstly, the Al–Ti–B master alloy was added in the melting process of these alloys, which promoted the heterogeneous nucleation during solidification. Only cellular dendrites with relatively little size can be observed even in the as-cast 7075 alloy. Secondly, the addition of Ni in the 7075 alloy decreased the grain size furthermore, and some equiaxed grains were found in the Ni–containing 7075 alloys from Figure 2. It suggested that the solidification behavior in Ni–containing 7075 alloys changed. According to the statistical results in Table 2, the initial solidification temperatures of Ni–containing alloys increased several degrees Celsius owing to the increase Ni content in 7075 alloys. The higher liquidus temperature for solidified alloy means greater undercooling during solidification process, which promoted nucleation. Thirdly, the growth behavior can be affected by the diffusion of solute elements in alloys. It can be seen from Figures 6–8 that the segregation content of intergranular Cu of the Ni–containing 7075 Al alloy was much higher than that of the 7075 Al alloy, and the gradient distribution of the Cu element from the intragranular to grain boundary was more obvious. The grain boundary segregation of Cu was similar to prior studies [34,35]; it was reported that Cu in a 7075 aluminum alloy resulted in an ordered parallel array in substitutional core sites [34]. In fact, there was a strong correlation among the concentrations of Mg, Zn, and Cu at the grain boundary for an Al–Zn–Mg–Cu alloy [35]. Considering the additional Ni in the 7075 alloy, it was analyzed that the activities of Cu and other alloy elements in the melt of the 7075 aluminum alloy decreased due to the interaction of nickel with other alloying elements, resulting in a decrease in the diffusion coefficient of alloy elements in the 7075 aluminum alloy melt. As a result, the grain growth process controlled by element diffusion slowed down.

Compared with the microstructure of convenient cast aluminum alloys, the as-cast 7075 alloy was composed of a large number of α–Al solid solutions surrounded by a small amount of eutectic structures. It is well known that solidification behavior seriously affects internal crack formation in the liquid–solid phase region [6]. According to the experimental

results, the additional 0.6% Ni in the 7075 Al alloy decreased intergranular discontinuities and shrinkage porosity from Figure 4, which may be partly attributed to the fine and equiaxed grains preformed prior to the eutectic solidification. Fine-equiaxed grains in the solidification process helped to decrease DCP temperature and enhanced the feeding ability. Meanwhile, the added Ni element in the 7075 alloy existed as an intergranular $Al_3Ni$ phase, and it was an allotropic eutectic phase formed after the primary phase grew continuously. What is more, the $Al_3Ni$ phase increased with the increase of Ni content, and it was formed as a eutectic phase in the final stage of solidification. Hence, the addition of Ni element was conducive to the release of stress and the reduction of intergranular discontinuities.

### 4.2. Effect of Ni Element on Solidification Behavior of Alloy

Figure 12 shows the results of thermodynamic calculations. The phases in the experimental alloys such as α(Al), Al–Cu–Mg phase, Fe–rich IMCs, $MgZn_2$, and $Al_3Ni$ had an obvious precipitation sequence, in which Fe–rich IMCs was firstly formed at the first stage of eutectic solidification, and Al–Cu–Mg and $MgZn_2$ phases formed at the last stage of the whole solidification. From Figure 12, the precipitation temperature of $Al_{13}Fe_4$ was higher than that of $Al_3Ni$ in 7075 Al–0.6 Ni alloy, while $Al_3Ni$ was formed in the 7075 Al–1.2 Ni alloy followed by the formation of α(Al) primary grains. The metastable phase formation has higher thermodynamic energy than the stable phase [36–38], and $Al_7Cu_2Fe$ preferred to form by Cu substitution of Al in $Fe_{13}Al_4$ with increasing Cu concentration [39]. Based on the thermodynamic calculations, $Al_7Cu_2Fe$ can be regarded as a kind of metastable Fe–rich phase, which formed prior to the $Fe_{13}Al_4$ in non-equilibrium solidification. $Al_2CuMg$ may have a similar solidification sequence with $Al_3CuMg$ in this alloy. In the eutectic formation temperature range, the calculated eutectic temperature of the 7075 Al–1.2 Ni alloy was higher by 5 °C than that of the 7075 Al–0.6 Ni alloy, which indicated that the increase of Ni content in the 7075 alloy led to the increase of the eutectic temperature. Accordingly, the solidification sequence of the as-cast 7075 in this study was determined as Liq → Liq + α–Al → Liq + α–Al + $Al_7Cu_2Fe$ → α–Al + $Al_2CuMg$ + $MgZn_2$, and the solidification sequence of the as-cast Ni–containing 7075 alloys was determined as Liq → Liq + α–Al → L + α–Al + $Al_7Cu_2Fe$ → L + α–Al + $Al_7Cu_2Fe$ + $Al_3Ni$ → α–Al + $Al_7Cu_2Fe$ + $Al_3Ni$ $Al_2CuMg$ + $MgZn_2$.

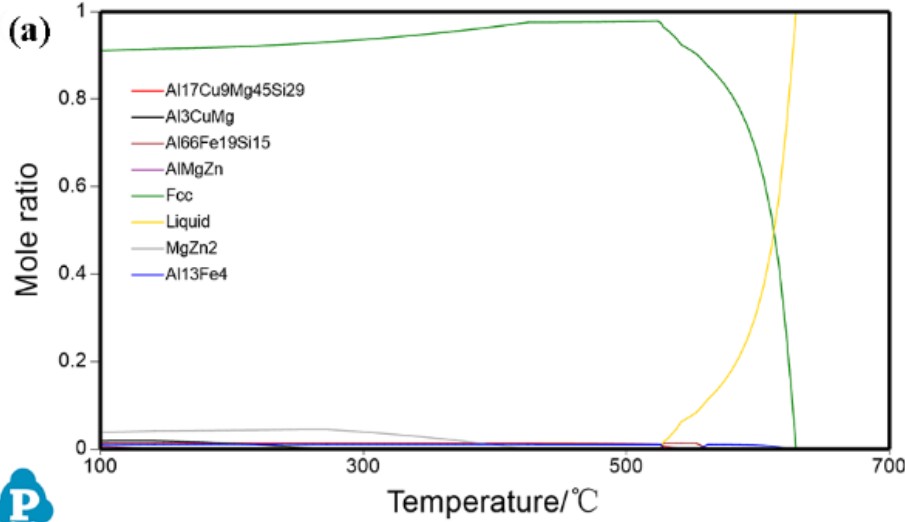

**Figure 12.** *Cont.*

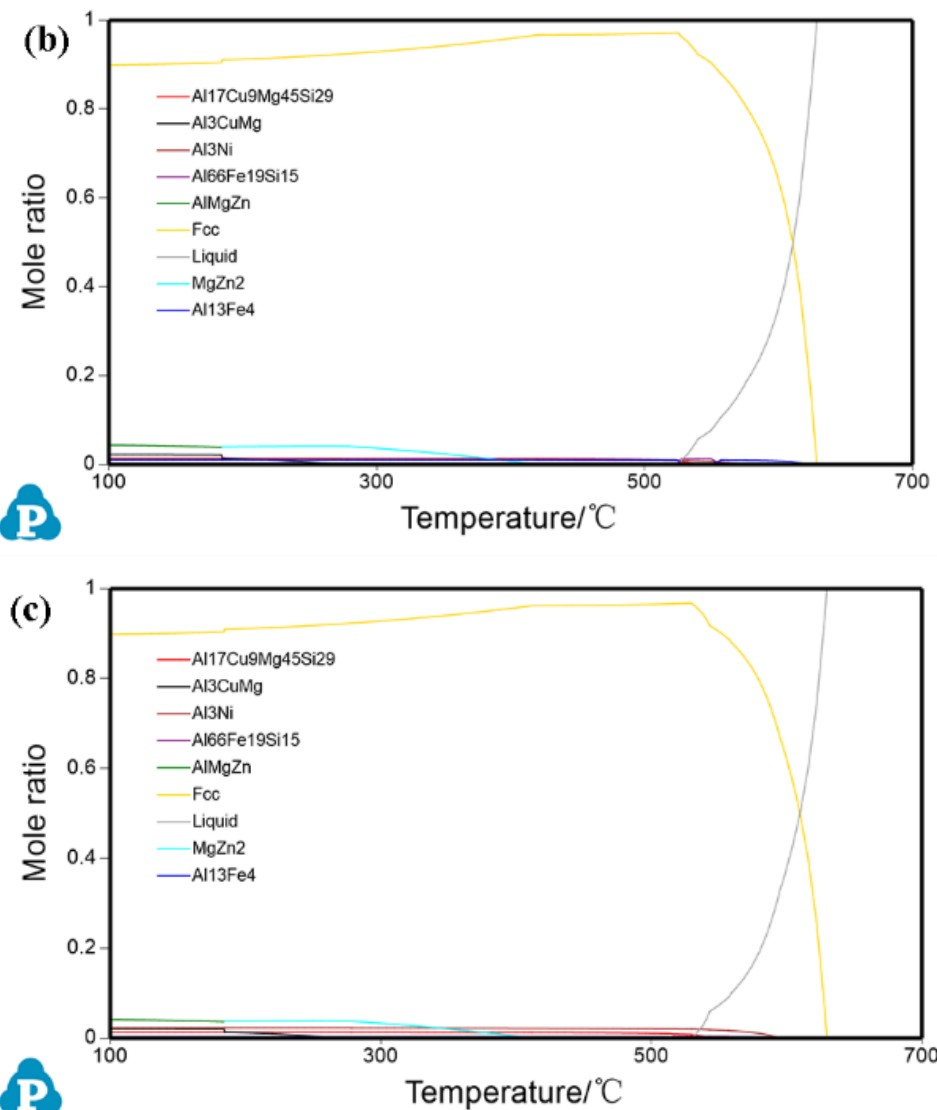

**Figure 12.** Phase fraction versus temperature (100–700 °C) of 7075 aluminum alloy with (**a**) 7075 Al, (**b**) 7075 Al–0.6 Ni, and (**c**) 7075 Al–1.2 Ni contents calculated by Pandat software.

The development of microstructure during alloy casting processes can be understood by nucleation and dendritic growth in cooling alloy melts. Compared with eutectic growth, the formation and early growth of eutectic nuclei are very difficult to observe [18]. Eutectic nucleation mainly depends on solidification conditions and alloy composition [40], which is mainly manifested in three modes [41]: (1) Nucleation by adhering to the tip of primary dendrite, (2) independent nucleation between primary dendrites, (3) nucleation in reverse thermal gradient direction.

The coupled lamellar eutectic phases in the as-cast 7075 aluminum alloy were caused by the growth behavior of the secondary $\alpha$(Al) phase. According to the adsorption process at the liquid/substrate interface of heterogeneous nucleation [18], a rough interface at the atomic level was formed between the $\alpha$(Al) and liquid phases during solidification. The rough interface formed a small plane through the adsorption of Al atoms, and then the $Al_2CuMg$ phase grew up through the spiral dislocation on the surface of Al phase. It can also be found from the cooling curve in Figure 9a that multiple peaks and troughs in the eutectic solidification stage were in response to the continuous nucleation and growth of coupled eutectic phases.

However, Ni–containing 7075 aluminum alloys consisted of a typical divorced eutectic structure. These intergranular phases in this investigation were identified as $Al_7Cu_2Fe$,

Al$_3$Ni, MgZn$_2$, and Al$_2$CuMg, and Fe–rich and Ni–rich IMCs such as Al$_7$Cu$_2$Fe and Al$_3$Ni were formed from residual liquids with high solute concentration due to the low solid solubility of Fe and Ni in Al. Combined with elements distribution of intergranular phases in Figures 6–8, it was suggested that the Al$_7$Cu$_2$Fe and Al$_3$Ni were firstly nucleated in the center of residual liquids and grew towards each other. Al$_2$CuMg was located between the primary phase and these two IMCs in this study. Combined with the thermodynamically calculated results shown in Figure 12, it was considered that Al$_2$CuMg attached to Al$_7$Cu$_2$Fe or Al$_3$Ni to nucleate, and its growth was controlled by the diffusion behavior of alloy elements. It can be seen from Figure 9b,c, that several troughs of cooling curves were related to the multiple IMCs phases evolution during solidification, which indicated that the solidification path changed when nickel was added in 7075 alloy.

### 4.3. Effect of Nickel on Mechanical Properties of As-Cast 7075 Alloy

The mechanical properties of the experimental alloys depended on the grain size, the morphology, and distribution of strengthening phases, as well as the characteristics of defects. Compared with 7075 aluminum alloy, Ni–containing 7075 aluminum alloys exhibited improved mechanical properties. Although the grain size of Ni–containing 7075 aluminum alloys decreased with the increase of nickel content, it did not achieve continuously increasing tensile strength and elongation when nickel content in the 7075 aluminum alloy was added up to 1.2%. The Al$_3$Ni phase had an extremely high tensile strength (216 MPa) and Young's modulus (116–152 GPa) [42], and these Al$_3$Ni particles were located at the intergranular region in nickel-containing 7075 alloys. It can be deduced that these Al$_3$Ni particles inhibited the movement of dislocations to a large extent so that Ni–containing 7075 aluminum alloys achieved high strength.

The intergranular structure in the as-cast 7075 Al alloy consisted of coupled eutectic phases, while the intergranular structure in as-cast Ni–containing 7075 aluminum alloys was made up of IMCs like Al$_7$Cu$_2$Fe, Al$_3$Ni, and Al$_2$CuMg. Owing to the differences in the amount and the formation mechanisms of intergranular phases, the as-cast 7075 aluminum alloy had more internal porosities than those in as-cast Ni–containing 7075 aluminum alloys. The higher the nickel content in the 7075 alloy was, the higher the amount that the intergranular Al$_3$Ni phase had. Although the Al$_3$Ni phase had a negative impact on the elongation, the decreased defects and refined grains were helpful for the increasing of the elongation. When the nickel content in the 7075 alloy increased from 0 to 0.6 wt%, both the elongation and strength were improved. While the nickel content in the 7075 alloy changed from 0.6 wt% to 1.2 wt%, the strength slightly decreased while the elongation basically remained unchanged. This may be attributed to the coarsening and aggregation of the strengthening Al$_3$Ni phase (as shown in Figure 4c).

From the fracture morphology in Figure 8, it can be seen that there were many inter-granular shrinkages in the 7075 Al–1.2 Ni aluminum alloy. According to the analysis of the intergranular shrinkage area of the fracture surface, there was also a positive correlation between the tensile strength of the alloy sample and the size of the intergranular shrinkage porosity area. The 7075 Al–0.6 Ni aluminum alloy had the highest strength among these three alloys, and it had the least intergranular shrinkages.

The tensile strength model of materials with holes can be expressed as follows:

$$\sigma = \sigma_0 (1 - p)^k \tag{1}$$

where $p$ is the porosity fraction, $k$ is the stress concentration factor, and $\sigma_0$ is the tensile strength at zero porosity, which depends on the mechanical properties of matrix and strengthening phase.

If the $k$ value of spherical porosity is 1, the tensile strength of the material is inversely proportional to the porosity content. It should be noted that even if the open porosity on fracture surface for the 7075 Al–1.2 Ni aluminum alloy increased significantly, its mechanical properties only decreased slightly. It was suggested that other microstructural factors affected the mechanical properties based on Equation (1). For Ni–containing 7075 alu-

minum alloys, the intergranular bonding strength between primary grains was enhanced by a eutectic mixture such as $Al_3Ni$, which improved the mechanical performance of the alloy matrix significantly. On the contrary, the continuous distribution of coarsening intergranular phases along the grain boundaries of the 7075 Al–1.2 Ni alloy also determined the intergranular mechanic behavior between primary phases because the fracture happened near the grain boundaries due to the stress concentration. It was analyzed that both the intergranular shrinkage defects and the bonding strength between primary grains affected the tensile properties of the experimental alloys.

## 5. Conclusions

The solidification microstructure of the 7075 alloy without and with different Ni contents and its influence on tensile properties were studied comparatively.

(1) The Ni addition in the 7075 alloy can refine the solidification grains and even cause equiaxed primary grains in the solidification, which enhanced the intergranular feeding ability. $Al_3Ni$ formed as an intergranular phase after the solidification of primary $\alpha$–Al grains, which increased the eutectic amount. Both the refined $\alpha$–Al grains and increased eutectic phase led to the decrease of intergranular porosities.

(2) The 7075 Al alloy solidified as a regular lamellar eutectic structure, while the intergranular phases in the 7075 Al–0.6 Ni alloy and the 7075 Al–1.2 Ni alloy consisted of multiple intermetallic compounds including $Al_3Ni$, $Al_7Cu_2Fe$, $MgZn_2$, and $Al_3CuMg$. $Al_3Ni$ solidified followed by the solidification of primary $\alpha$–Al and $Al_7Cu_2Fe$. Finally, the $Al_2CuMg$ phase and $MgZn_2$ phase were sequentially precipitated from the residual liquid.

(3) The as-cast Ni–containing 7075 alloys showed enhanced tensile strength and elongation compared with the as-cast 7075 alloy. In addition, 0.6% Ni addition in the 7075 alloy can get optimal mechanical properties. Too much Ni addition produced coarsening $Al_3Ni$ phase and led to premature brittle fracture.

**Author Contributions:** Conceptualization, K.W., N.H. and F.L.; methodology, K.W., N.H. and L.W., Writing—original draft, K.W., H.Q. and L.W.; Writing—review and editing, K.W., H.Q. and N.H.; Investigation, S.M., H.Q. and L.W.; Software, L.W. All authors have read and agreed to the published version of the manuscript.

**Funding:** This research was funded by the State Key Laboratory of Solidification Processing in NPU, (Grant No. SKLSP201910) and the National Natural Science Foundation of China (Grant No. 51974050).

**Data Availability Statement:** The data presented in this study are available in the article.

**Acknowledgments:** We would like to thank Yang ZHOU for their assistance with EMPA analysis.

**Conflicts of Interest:** The authors declare no conflict of interest.

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
