# Peer review of "Effect of Nickel Addition on Solidification Microstructure and Tensile Properties of Cast 7075 Aluminum Alloy"

_crystals, doi:10.3390/cryst13111589_

Round 1

Reviewer 1 Report

Comments and Suggestions for Authors

This study investigated the effect of adding nickel to the 7075 Al alloy, the quality of the work is good but the problem is the topic, if you search on the net you can find a lot of papers on this topic so the novelty of this work still is hidden for me as a reader. however, I suggest some modifications in order to make it better for readers:

1. the authors need to extend the introduction, the order can be like this:

  1. Background and Context

    • Introduction to Casting 7075 Aluminum Alloy as a widely used material in various industries.
    • Highlight the importance of understanding its microstructure and tensile properties for engineering applications.
  2. Significance of Alloying Elements

    • Explanation of the role of alloying elements in modifying material properties.
    • Specific focus on the role of nickel as a critical alloying element in aluminum alloys.
  3. Research Gap

    • Identification of the existing gap in the understanding of how nickel influences the microstructure and tensile properties of Casting 7075 Aluminum Alloy.
  4. Objective of the Study

    • Clear statement of the study's primary objective: to investigate the influence of nickel content on the microstructure and tensile properties of Casting 7075 Aluminum Alloy.
  5. Scope of the Research

    • Briefly mention the methodology and parameters under investigation.
    • Set the boundaries for the study by defining the range of nickel content considered.
  6. Importance of Findings

    • Discuss the potential impact of the study's findings on material selection, design, and applications in industries where Casting 7075 Aluminum Alloy is utilized. you can use this as reference: https://doi.org/10.3390/mi14051081 
  7. Structure of the Paper

    • Provide an overview of how the article is organized, including sections to be covered, such as methodology, experimental setup, results, and discussion.

2. Check the typography errors inside the body text for example: impac2ts or casing etc...

3. The three types of aluminum alloy samples have the same melting conditions, you need to mention this condition.

4. For all equipments you must mention the model , for example: 15 KW electric resistance furnace (model, company, city, country), ZEISS Axioskop optical microscope, etc...

5. what is this: PanPhaseDiagram , software or method, you need to explain it with reference

6.make whole the manuscript homogenous , figure or fig. ???? make then similar in the manuscript.

7. The colors red and yellow are not good choice for the SEM images , change them to green and blue.

8. Al-20wt%Ni / 7075Al-1.2%N / 7075Al-1.2Ni / in the whole manuscript you wrote them as you want???!?!? 

9. It would be better if you draw a table with the thermal / physical properties of the Al used in this study.

10. Figure 2. b and c , the authors must explain the diffrences between two SEM images, this is the point.

11. Adding more Ni, how the intensity of the XRd peak reduced?

12. Check all the units again some of the is not correct: 187Mpa?? MPa

13. Figure 9, 10 you should separate all images label them, and mention them in the caption.

14. For figure 12 check these references: https://doi.org/10.1016/j.msea.2019.04.018 https://doi.org/10.3390/mi14051081 https://doi.org/10.3390/jcs7070300 

Comments on the Quality of English Language

minor English language editing needs

Author Response

Thanks for your precious comments on this manuscript, we revised this draft according to you suggestions and comments.  thanks a lot.  

Reviewer 2 Report

Comments and Suggestions for Authors

The manuscript studies the effect of nickel on the microstructure and tensile properties of casting 7075 Aluminum Alloy. The manuscript has serious flaws and cannot be recommended for publication in its current status:

1- The English language of the manuscript needs improvement by an English native speaker.

2- Regarding the addition of Zr, Sc and Cr (in the current work) to Al-Zn-Mg-Cu alloy, please explain in detail in the introduction the effect on the microstructure. Please cite the following reference for clarifying this point and explain the reason for choosing the current chemical composition for your study.

https://doi.org/10.1016/j.corsci.2021.109895

3- Regarding the stoichiometry of Al-Fe insoluble particles, please explain why you reported the current one and compare them with the references.

4- We usually see the formation of S or S' phases with Al2CuMg composition.

https://doi.org/10.3390/ma16124384

Please compare the current precipitates with the reference and explain what is the kinetics of the phase precipitation.

5- Please explain the kinetics of the phase transformation reported, e.g. Al13Fe4→Al3Ni.

6- Please also explain the segregation of the elements in the alloy and the effect of the nickel addition to the alloy. Please also use the following references.

https://doi.org/10.1016/j.vacuum.2020.109937

https://doi.org/10.1002/pssa.201800240

Comments on the Quality of English Language

The English language of the manuscript needs improvement by an English native speaker.

Author Response

Dear reviewer, Thanks for your precious comments on this manuscript, we revised this draft according to you suggestions and comments.  thanks a lot.  

Reviewer 3 Report

Comments and Suggestions for Authors

Please see my comments/suggestions hereafter:

1. Title: You mention casting, but the designation of the alloy is for a wrought alloy. I would advise that you either use the composition of the alloy or that of the cast alloy.   

2. Introduction: How does hot cracking during casting relates to tensile tests after solidification?   

3. Introduction: The novelty of this work should be better highlighted. The added value of this article is only described in this sentence ‘However, the relationship between eutectic phases and intergranular discontinuities, as well as the microscopic interaction between them, are still lacking’. Since there are a lot of articles published on the effect of Ni on the mechanical properties of Al alloys you should clearly state and discuss what is the new contribution.

4. Experimental: You mention that ‘Casting samples of 7075 aluminum alloy without nickel and with 0.6% and 1.2% nickel were prepared’. Is that at.% or wt.% ?

5. Table 1 and 2: Spread of composition should be mentioned.

6. Experimental: Was the casting performed under inert atmosphere? If not did you use any Fluxes and Degas Tablets?

7. Experimental: Due to the solidification process you have different solidification rates and thus microstructure from the surface layers to the bulk. Location from where you obtained the samples (tensile and metallographic) should be clarified.

8. Experimental: After cutting the samples from the ingot, did you perform any post treatments and annealing to remove stresses from processing?

9. Experimental: According to ASTM D628, the thickness of the samples should be much smaller that the length to have uniaxial stresses. How and why did you select this sample geometry?

10. Experimental: Grinding, polishing and resulting roughness should be described.

11. Experimental: XRD analysis you should mention the filter and monochromator.

12. Results: The addition of Ni results in a further decrease of the grain size of the alloy. Please provide location of measurement and quantify this decrease.

13. Results: Indeed, the addition of Ni in this range does not change the formed phases as shown in the XRD spectograph, but what about the ratio between the peaks indicating volumetric changes within the alloy?

14. Results: Did you used any image analysis software (e.g. Image Analysis) to quantify amount of intermetallics per area?

15. Results: How repeatable are your cooling curves?

16. Results: Aluminium alloys exhibiting brittle failure, this contradicts existing literature. Why is that?

17. EPMA measurements should be moved to results or the results should ne merged with the discussion as Results and discussion.

18. Discussion: How did you investigate the release of stresses. No significant shifts can be seen in the XRD peaks, indicating no major differences in macro-strain between the samples.

19. Discussion: Explanation of mechanical properties and link to microstructure is not clear. The following factors that play a role namely, grain size, intermatellics (composition, distribution and volume) and defects (porosity, cracks etc.) and residual stresses (which you did not took into account). The synergism between those factors determines the outcome of your tests. This should be made clear in your discussion by quantifying your results (e.g. % of intermetallics and pores, reduction of grain size etc.) and comparing them.    

20. Throughout there are mistakes e.g.: (Abstract) impac2ts impacts,  (Experimental) wt% wt.%, (Results) 187Mpa 187 MPa, (Discussion) At last, → ? etc.

Comments on the Quality of English Language

Throughout there are mistakes. The manuscript needs editing.  

Author Response

(The authors gave the same response as above.)

Reviewer 4 Report

Comments and Suggestions for Authors

The authors presented an article «Effect of Nickel on Microstructure and Tensile Properties of Casting 7075 Aluminum Alloy». The manuscript is clear, relevant to the field and presented in a well-structured manner. The authors are advised to consider the following comments for this paper.

·         Abstract

The abstract need to be improved. Please provide the main quantitative and qualitative research core findings. Demonstrate in the abstract novelty, practical significance. Briefly list the input and output parameters of the research.

·         Introduction

Seemingly, a literature review was given. However, they were just summarized one- by-one. The authors have to stop after writing each example and think about the contributions and lack of knowledge for each paper. After that, in the final lines of the introduction give the blank spots of the topic. Then it will be clear what did authors make differently from the open literature. Some key references were studied in several paper as follows:  (I) Influences of Cr on the microstructural, wear and mechanical performance of high-chromium white cast iron grinding balls, (II) Experimental investigation of tensile strength and thermal conductivity of nanoparticle reinforcement composite materials,

In the last paragraph of the introduction section; What is the scientific novelty of the paper? What is the practical value? What makes this approach different from other researchers? Please specify. Gap and significance of the work must be included.

·         2 Experimental methods

Why did you choose this material? Please specify.

What are the standards used in the tests?

Please provide more detailed basis and reference for selecting Ni mass fraction and their levels (0.6 and 1.2). Please specify.

·         3. Result

It is useful to add explanations of parameters to the results obtained. At least four-five sentences for each Figures. The results obtained should be explained by supporting the literature.

The phases can be shown on Figure 2.

Arrows and circles on Figure 4 can be given more clearly.

Figure 7 should be discussed in detail. Figure 7 is difficult to understand. Sample names can be written on the X axis.

How many repetitions were performed.

·         Conclusions

The conclusions need to be improved. The results are written long. It is necessary to more clearly show the novelty of the article. Add qualitative and quantitative results of your work. What is the difference from previous work in this area? Show practical relevance. What are the differences from previous works?

Suggestions should be made to increase the studies to be done in this field in conclusion section.

·         Authors should carefully study the comments and make improvements to the article step by step. All changes should be highlighted in color.

Author Response

(The authors gave the same response as above.)

Round 2

Reviewer 1 Report

Comments and Suggestions for Authors

1. Figure 2. Optical microstructure of 7075 aluminum alloy????? why you bring this up when all of them are almost similar??? Are you sure that they are optical?? not SEM??

2.Title is not good, make it more scientifical, there is a lot of articles with this title???!?!?!??!?! add some more scientifical ....

3. remove from Keywords Microstructure and replace it with another word

4.in the first paragraph in the introduction why is it bold?? not justify?

5.what is this point?

6. it seems somebody wrote this manuscript which doen't know a lot about materials science or mechanical eng. what you mean Tensile test pieces ???? Tensile test specimens...

7.  The 137: grain boundary intersection of cast 7075 alloy was determined as 58 ± 4.1 μ m.  how you calculated it?

8.how the authors defined it?? its need more magnification on the curves!

9. figure 5, do you have EDX analysis or EBSD????

10. in the previous comment i mentioned , hemogenous?? fig or figure?? in the bodytext or caption similar.

11. Table 2, if you dont have the elements why do you mentioned them?

12. what isn this sign?

13.conclusion must be extended....it very short by the results....

14. new references must be added, where are the references which you added? please highlight them???!?!?

Author Response

Dear reviewer,

      We have revised and highlight in the revised manuscript. thanks for your suggestion. 

Best regards

Kai WANG

Reviewer 2 Report

Comments and Suggestions for Authors

The manuscript seems acceptable after extensive revisions.

Comments on the Quality of English Language

The English language is ok and needs a slight revision and polishing.

Author Response

Dear Reviewer,

     Thanks for your comments, it,s very important for us to improve the quality and research level. we have revised this manuscript.

Bset regards

kai wang

Reviewer 3 Report

Comments and Suggestions for Authors

After reading the updated version and your reply to my comments, I believe that the manuscript has improved. However, a few points still remain. See my comments hereafter: 

1. You mention that 'The formation of Al3Ni in Ni-containing 7075 aluminum alloys can be attributed to a relative low Ni content in 7075 alloy' → What do you mean by relative low? If you form intermetallics it means that Ni is above the solubility limit and if you detect it by XRD the intermetallic phase should be at least 2-3 %.

2. You mentioned that the discussion on XRD has been revised, but the addition made within the text are very limited. As mentioned previously you should discuss if there are any changes in the ratio and/or shifts of peaks.

 3. Again there are grammatical mistakes within the text. Please correct.  

Comments on the Quality of English Language

Minor editing of English language required.

Author Response

Dear revirewer, 

      Thanks for you comments on our manuscript, and we revised those. 

best regards

kai wang

Round 3

Reviewer 1 Report

Comments and Suggestions for Authors

1. remove this from abstract: This study demonstrated that the designation of eutectic composition and eutectic structure are effective technical methods to improve the mechanical properties of cast high-strength aluminum alloy.

2. casing process or casting process?

3.how is the volume the triangular mold?

4. figure 1, check the caption??

5.cite this ASTM E8 / E8M?? ref?ASTM E112-12?

6. for all equipment you should say the model; manufacturer , JEOL JXA-8530F plus electron probe microanalysis

7.the range of youngs modulus is very different 116-152GP?? are you sure?

8. still references are not all relevant.

Author Response

Dear, 

       thanks for you comments, and we revised this manuscrita again. thanks for you patiance.

Best regrads

kai wang
